# Psychosocial and Health Effects of the COVID-19 Pandemic Experienced by Staff Employed in Social Welfare Facilities in Poland and Spain

**DOI:** 10.3390/ijerph20043336

**Published:** 2023-02-14

**Authors:** Edyta Janus, Raquel Cantero Téllez, Katarzyna Filar-Mierzwa, Paulina Aleksander-Szymanowicz, Aneta Bac

**Affiliations:** 1Faculty of Motor Rehabilitation, University of Physical Education, 31-571 Krakow, Poland; 2Faculty of Health Sciences, University of Malaga, 29071 Málaga, Spain

**Keywords:** social welfare facilities, COVID-19, pandemic, psychosocial effects

## Abstract

The COVID-19 pandemic had a very significant negative impact on the physical and mental health of various professional groups. Therefore, the aim of this study was to assess the psychosocial and health effects of the COVID-19 pandemic experienced by staff employed in social welfare institutions in Poland and Spain. The study involved 407 people, including 207 from Poland and 200 from Spain (346 women and 61 men), working in social care facilities. The research tool was the authors’ questionnaire consisting of 23 closed-ended, single- or multiple-choice questions. The study has indicated that the COVID-19 pandemic had negative health and psychosocial effects on employees of social welfare facilities. In addition, it has been shown that the severity of the psychosocial and health effects of the COVID-19 pandemic differed between the countries studied. Employees from Spain statistically significantly more often declared deterioration in most of the surveyed indicators, except for mood deterioration, which was experienced more by employees from Poland than their peers from Spain.

## 1. Introduction

In December 2019, the Chinese office of the World Health Organization (WHO) was notified of cases of pneumonia of unknown aetiology occurring in Wuhan [1]. This was coronavirus disease 2019 (COVID-19), a highly contagious disease with a long incubation period, attacking the human respiratory system [2]. Owing to the rapid spread of the virus around the world, the WHO announced a pandemic on 11 March 2020. The first case of COVID-19 infection in Poland was recorded on 4 March 2020 [3] and the first public restrictions were introduced on 13 March 2020, when the Polish government announced the Risk of Pandemic Status (Journal of Laws No. 2019.1239) [4], followed by the Pandemic Status on 20 March 2020 (Regulation of the Minister of Health of 20 March 2020, Journal of Laws No. 491) [5]. In Spain, the first case of COVID-19 was recorded on 31 January 2020, but it was not until 12 March that the country took significant measures, including social restrictions [6]. On 14 March, a state of emergency was declared in Spain by Royal Decree (463/2020) [7]. The course of the pandemic was different in these two countries. From 4 April 2020, Spain was the second highest country in the world in terms of the number of infected people and the second highest in terms of the number of deaths [6], while the number of cases of and deaths from COVID-19 infections in Poland were at an average level [8].

The social restrictions resulting from the pandemic were similar in both countries. They affected practically every sector of the state, protecting the citizens working in them, but, for obvious reasons, they could not fully cover health or social care facilities. Due to the uniqueness of the situation and the rapid development of the pandemic, the health sector was overloaded in both Poland and Spain. Its employees were not only the most exposed to COVID-19 infection but also felt the psychosocial burden related to the development of the epidemic the most [9]. Psychosocial risks, which may cause mental, social or physical damage to employees through the mechanism of stress [10], and the physical burden resulting from virus infection caused very adverse long-term effects in this social group [11,12,13].

The psychological effects of COVID-19 in different professions have been studied in recent years in some countries to understand how the pandemic has affected workers, especially in the health area [14,15,16], but no specific results for social welfare institutions have been reported. Instead, other social factors specific to each country could interfere with the results of these studies.

Various types of social institutions became special cases during this difficult time. In both Poland and Spain, the staff working in them carried the same burden as other healthcare professionals but did not receive the same support as hospitals [17]. We have not found previous studies where the same survey was distributed among social welfare institutions from different countries. As there are no reports in the available literature about the differences between countries on the negative effects of the COVID-19 pandemic in this professional group, the aim of this study was to assess the psychosocial and health effects of the COVID-19 pandemic experienced by staff employed in social welfare institutions in Poland and Spain to check significant differences. Based on the aim of the study we hypothesized:Employees from Spain have more concerns about their own health than employees from Poland.Employees from Spain, more often than employees from Poland, make more changes in their work.Employees from Spain make changes in the sphere of social relations more often than employees from Poland.Employees from Spain have a worse mood than employees from Poland.

We presume that our hypotheses will be confirmed, based on the fact that the pandemic in Spain had a much more severe course than in Poland.

## 2. Methods 

### 2.1. Participants 

The study involved 407 people, including 207 from Poland and 200 from Spain. Employees of social welfare institutions located in Poland and Spain were selected using purposive sampling. The study included social welfare institutions operating in the public area and providing 24-h care. The purpose of those institutions is to meet living, educational, and social needs in accordance with legal regulations in a given country. All approached participants agreed to participate in the study and gave written, informed consent. The respondents included 346 women and 61 men. Gender inequality results from the fact that, in both Poland and Spain, professions related to social care are feminised. The employment structure of the respondents was as follows: physicians (*N* = 5, 1.2%), psychologists (*N* = 25, 6.1%), nurses (*N* = 19, 4.7%), social workers (*N* = 27, 6.6%), occupational therapists (*N* = 142, 34.9%), physiotherapists (*N* = 44, 10.8%), carers (a person who provides care and support services to the ward) (*N* = 95, 23.4%), and other (instructors, art therapists and music therapists) (*N* = 50, 12.3%) (see Table 1).

### 2.2. Instruments

The questionnaire was built for the present study by the authors. An item-based questionnaire, Spanish and Polish version, was constructed following a comprehensive literature search. To maintain equality between the two questionnaires, both versions were back translated by a translator and, subsequently, reviewed by the research team. In this way, the accuracy of the translation was verified. The questionnaire consisted of 23 closed-ended, single- or multiple-choice questions. The questions included in the research tool concerned the activities of respondents during the pandemic, changes in working time, professional tasks performed, interpersonal relations, and issues related to the area of mental functioning and health, understood in a broad sense. At the end of the questionnaire, there was a data sheet containing basic sociodemographic data.

### 2.3. Statistical Analysis

All the results obtained were analysed using the Statistica software (StatSoft, Hamburg, Germany, v25). It was applied to perform the Mann-Whitney *U* tests, Kruskal–Wallis tests, Fisher’s exact tests and chi-squared tests. The significance level was *p* = 0.05.

## 3. Results

The first examined difference was that between the respondents’ country of residence and their attitude towards their own health. There were only two statistically significant differences between the variables studied. Respondents from Spain more often indicated that, due to the COVID 19 pandemic, they constantly listened to their bodies, looking for any symptoms of infection, and less often took probiotics and protective drugs compared to the respondents from Poland. In the case of other variables, there were no statistically significant changes (see Table 2).

Another analysed difference was that between the respondents’ country of residence and the reported need to modify particular forms of job-related activity due to COVID-19. The difference between the countries was statistically significant, as respondents from Spain more often indicated that they had to change particular forms of job-related activity due to the pandemic (see Table 3).

The respondents were also asked what job-related activities they had to change due to the COVID-19 pandemic. Employees from Spain statistically more often worked remotely, took part in online team meetings and in online supervision. For the remaining variables—an increase or decrease in job-related duties and problems with self-acceptance—no statistically significant differences were noted (see Table 4).

Another examined difference was that between the respondents’ country of residence and changes in the sphere of social relations due to the COVID-19 pandemic. A statistically significant difference was noted between the surveyed countries. Namely, employees from Spain declared a greater range of changes in social relations than employees from Poland and this difference was statistically significant (see Table 5).

The last difference studied was that between the respondents’ country of residence and changes in mood due to the COVID-19 pandemic. The respondents from Spain declared a clear improvement in mood and the respondents from Poland a clear deterioration. The difference was statistically significant (see Table 6).

## 4. Discussion

In the available literature, the health and psychosocial effects of the COVID-19 pandemic are widely discussed, but these studies mainly concern doctors and nurses working in hospitals [11,12,18,19,20,21,22]. However, there are no reports on people working in social welfare institutions. Stocchetti et al. [20] examined 136 employees of intensive care units during the second wave of COVID-19. After analysing four questionnaires, the authors found that the most frequently reported negative effects of the COVID-19 pandemic were depression, anger, insomnia, and occupational burnout. According to Nowicki et al. [18], who surveyed 325 nurses, this professional group also had symptoms of post-traumatic stress, a reduced sense of security, and increased meaning of life. Preti et al. [19] additionally pointed to problems related to a generally poor mental state and fears of infection among doctors and nurses. Pollock et al. [23] have also shown that, in addition to anger, depression, and stress, health and social care workers also reported cognitive and social disorders that could have had a negative impact on their professional work. Our study concerned the staff of social care facilities; the respondents were not only doctors or nurses but also physiotherapists, occupational therapists, psychologists, and social workers, among others. Responding to the questions in the survey, respondents pointed to changes in mood resulting from the COVID-19 pandemic and changes in attitudes towards their own health.

Despite the relatively short observation period, the available literature contains various reports on the effects of the COVID-19 pandemic [13,23]. Although these reports focus mainly on medical personnel, their authors use various questionnaires in their research. Such a heterogeneity of assessment tools, and the fact that a significant number of the available questionnaires evaluating psychosocial effects are not validated in either Poland or Spain, made the authors of this study decide to use a questionnaire of their own design in the research. Our own survey allowed us to make comparisons in two different countries, focusing on the exact variables we considered most relevant to our project, which distinguishes our study from the reports of other authors. For our study, we selected two European Union countries that are distant from each other, and where the course of the pandemic differed in intensity. Like people in other countries of southern Europe, such as Italy, Spaniards showed low resistance to COVID-19 infections, which translated not only into the number of cases among citizens but also into an extremely high mortality rate compared to Polish residents. Such a course of the epidemic could be the reason that the surveyed employees from Spain, statistically significantly more often than employees from Poland, indicated changes in attitudes to their own health, which involved mainly listening to their bodies constantly and looking for any symptoms. Those findings partially supported the hypothesis that employees from Spain have more concerns about their own health than employees from Poland. On the other hand, employees from Poland statistically significantly more often used prophylaxis, in the form of protective drugs or probiotics. This situation is only seemingly paradoxical. Poland is a country with a harsher and cooler climate than Spain and, therefore, Poles are used to seasonal epidemics of infectious diseases (including influenza) and the associated increased use of drugs, vitamins, supplements, and probiotics. The more severe course of the pandemic in Spain than in Poland could also have had an impact on significantly more frequent changes in social relations reported by the surveyed employees from Spain.

The surveyed respondents from both countries were also asked about the reported need to change their job-related activities and what types of changes were made. In this case, the professional lives of employees from Spain were more significantly affected by the pandemic, but also the employees from Spain used modern work techniques and technologies significantly more than employees from Poland (remote work, training, and online meetings). These results support the hypothesis that employees from Spain, more often than employees from Poland, make more changes in their work. This situation may be explained by the fact that Spain is a richer country than Poland (as confirmed by both gross domestic product and actual individual consumption indicators), which may have allowed for more effective equipping of social welfare facilities with at least appropriate computer equipment.

The next question asked of respondents concerned changes in the sphere of social relations due to the COVID-19 pandemic. Social relations are a very important element affecting the level of well-being of the community, especially life satisfaction. [24,25]. Some authors report that the COVID-19 pandemic, due to inter alia forced isolation at home or fear of contracting a disease, could cause significant changes in social relationships, adversely affecting the quality of life [26]. In our research, employees from Spain declared greater changes in social relations compared to employees from Poland, which supported the hypothesis. This situation may be explained by the fact that the COVID-19 pandemic was more severe in Spain than in Poland.

The last question asked of respondents concerned changes in mood due to the COVID-19 pandemic. Although the pandemic affected Spain much more, Spanish employees of social care facilities much more often indicated an improvement in their mood. Those results do not support the hypothesis that employees from Spain have a worse mood than employees from Poland. While our study confirms reports from the literature in the case of employees from Poland, this result is surprising in the case of employees from Spain because the existing reports on changes in mood among employees from Spain indicate its deterioration during the COVID-19 pandemic [27,28].

Considering that our article is one of the first to describe the psychosocial effects of the COVID-19 pandemic among employees of social welfare facilities and, to the best of our knowledge, one of the first to compare two countries, we recommend further research in this group of health professionals and exploring the topic of similarities and differences between individual countries, which may allow for a better response to global medical emergencies in the future. Instead, future research should include not only a specific questionnaire but also complete the questionnaire results with specific depression or anxiety scales in order to reach powerful conclusions.

## 5. Study Limitations

This study is not without limitations. Good guidelines for future studies include sampling representatives of the same profession and level of education, and selecting an equal number of respondents from individual occupational groups. This may be important owing to different levels of remuneration received by these employees, depending on the level and type of education, which may affect the answers provided in the survey. Another limit is not considering the differences in the attitudes about vaccination of workers from Spain versus Poland. There are significant differences in attitudes about vaccination [29,30] that may explain the differences in mood deterioration. Further studies should investigate the attitudes about vaccination and vaccination status as well. The authors are also aware that differences between countries may have been influenced by factors, such as age differences (younger respondents from Spain) or level of education (more respondents from Spain had higher education). At the end, the authors wanted to emphasize that they are aware of the unequal gender distribution in this research. However, this gender inequality results from the fact that, in both Poland and Spain, professions related to social care are feminised.

## 6. Conclusions

Our study showed that the COVID-19 pandemic had negative health and psychosocial effects on employees of social welfare facilities. In addition, it indicates that the psychosocial and health effects of the COVID-19 pandemic differed in severity between the countries surveyed. Employees from Spain statistically significantly more often declared deterioration in most of the surveyed indicators, with the exception of mood. Employees from Poland indicated significant mood deterioration, in comparison to their colleagues from Spain.

## Figures and Tables

**Table 1 ijerph-20-03336-t001:** Demographic data.

		Poland		Spain		Total	
		*N*	%	*N*	%	*N*	%
Gender	Women	181	87.5	165	82.5	346	85
	Men	26	12.5	35	17.5	61	15
Age (yrs)	<20	0	0	23	11.5	23	5.5
	21–30	28	13.5	92	46	120	29.5
	31–40	83	40.1	23	11.5	106	26
	41–50	72	34.8	35	17.5	107	26.5
	51–60	19	9.2	24	12	43	10.5
	61–70	5	2.4	2	1	7	1.7
	>70	0	0	1	0.5	1	0.3
Education	vocational	4	1.9	7	3.5	11	2.7
	secondary	74	35.8	26	13	100	24.5
	higher	129	62.3	167	83.5	296	72.8

**Table 2 ijerph-20-03336-t002:** Respondents’ country of residence and changes in their attitudes towards their own health.

			Poland	Spain	Differences between the Countries
I constantly listen to my body, looking for any symptoms of infection	no	*N*	189	150	*p <* 0.001
		%	91.30%	75.00%	
	yes	*N*	18	50	
		%	8.70%	25.00%	
I try my best to protect myself from contracting infection	no	*N*	96	77	*p =* 0.108
		%	46.40%	38.50%	
	yes	*N*	111	123	
		%	53.60%	61.50%	
I take probiotics and protective drugs prophylactically	no	*N*	162	191	*p <* 0.001
		%	78.30%	95.50%	
	yes	*N*	45	9	
		%	21.70%	4.50%	
I try to visit specialist doctors regularly	no	*N*	197	193	*p =* 0.502
		%	95.20%	96.50%	
	yes	*N*	10	7	
		%	4.80%	3.50%	
I have done diagnostic tests that I have not done before	no	*N*	193	194	*p =* 0.079
		%	93.20%	97.00%	
	yes	*N*	14	6	
		%	6.80%	3.00%	
Other	no	*N*	198	195	*p =* 0.307
		%	95.70%	97.50%	
	yes	*N*	9	5	
		%	4.30%	2.50%	

**Table 3 ijerph-20-03336-t003:** Respondents’ country of residence and indications regarding the need to modify particular forms of job-related activity due to COVID-19.

			Poland	Spain	Differences between the Countries
I need to change particular forms of job-related activity due to COVID-19 pandemic	yes	*N*	102	165	*p <* 0.001
		%	49.30%	82.50%	
	no	*N*	105	35	
		%	50.70%	17.50%	

**Table 4 ijerph-20-03336-t004:** Respondents’ country of residence and the type of job-related activities changed.

				Poland	Spain	Differences between the Countries
Type of job-related activity changed due to the COVID-19 pandemic	Remote work	no	*N*	177	143	*p* < 0.001
			%	85.50%	71.50%	
		yes	*N*	30	57	
			%	14.50%	28.50%	
	Participation in online employee team meetings	no	*N*	190	115	*p* < 0.001
			%	91.80%	57.50%	
		yes	*N*	17	85	
			%	8.20%	42.50%	
	Participation in online supervision	no	*N*	198	162	*p* < 0.001
			%	95.70%	81.00%	
		yes	*N*	9	38	
			%	4.30%	19.00%	
	Increase in job-related duties	no	*N*	122	118	*p* = 0.990
			%	58.90%	59.00%	
		yes	*N*	85	82	
			%	41.10%	41.00%	
	Decrease in job-related duties	no	*N*	194	179	*p* = 0.124
			%	93.70%	89.50%	
		yes	*N*	13	21	
			%	6.30%	10.50%	
	Problems with self-acceptance	no	*N*	205	191	*p* = 0.028
			%	99.00%	95.50%	
		yes	*N*	2	9	
			%	1.00%	4.50%	

**Table 5 ijerph-20-03336-t005:** Respondents’ country of residence and changes in the sphere of social relations.

			Poland	Spain	Differences between the Countries
Changes in the sphere of social relations due to the COVID-19 pandemic	They definitely changed	*N*	53	118	*p* < 0.001
		%	25.60%	59.90%	
	They somewhat changed	*N*	86	56	
		%	41.50%	28.40%	
	They remained unchanged	*N*	52	22	
		%	25.10%	11.20%	
	They hardly changed	*N*	16	0	
		%	7.70%	0.00%	
	They definitely did not change	*N*	0	1	
		%	0.00%	0.50%	

**Table 6 ijerph-20-03336-t006:** Respondents’ country of residence and changes in mood.

			Poland	Spain	Differences between the Countries
Changes in mood due to the COVID-19 pandemic	It definitely improved	*N*	1	175	*p* < 0.001
		%	0.50%	87.90%	
	It somewhat improved	*N*	1	24	
		%	0.50%	12.10%	
	It did not change	*N*	39	0	
		%	20.60%	0.00%	
	It somewhat deteriorated	*N*	115	0	
		%	60.80%	0.00%	
	It definitely deteriorated	*N*	33	0	
		%	17.50%	0.00%	

## Data Availability

The data presented in this study are available on request from the corresponding author.

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
