# Peer review of "Psychosocial and Health Effects of the COVID-19 Pandemic Experienced by Staff Employed in Social Welfare Facilities in Poland and Spain"

_ijerph, 2023, doi:10.3390/ijerph20043336_

Round 1

Reviewer 1 Report

The aim of this study is very interesting because focus on the psycho-social and health effects of the COVID-19 pandemic experienced by staff employed in social welfare institutions and its comparison between two European countries, given that, as authors indicated, there are few reports on people working in this field.

However, some suggestions are indicated below in order to improve some sections of the manuscript:

- It would be more appropriate to include the method section after the introduction, which should include the following sections:

Methods:

Subjects/Participants

Assessment/Instruments

Procedures/Statistical analysis

Also, ethical considerations should also be indicated.

- It is interesting to have created a questionnaire by the researchers, however, it would have been interesting to have also administered some other standardized questionnaire to have a more scientifically consistent assessment, for example, the evaluation of anxiety, depression, stress or other another variable.

- Since a questionnaire has been created by the researchers, I suggest to include the complete questionnaire in the paper or explain better the parts of it in the method.

- It is important to specify more precisely the assessment period, including the years, in order to know more precisely the moment of the time of the evaluation.

- Try to include some more recent references regarding this field.

Author Response

Reviewer 1

Dear Reviewer,

We would like to thank you for a detailed review of our manuscript and all of your valuable remarks. We have addressed them all in detail below.

Reviewer: It would be more appropriate to include the method section after the introduction, which should include the following sections:

Methods:

Subjects/Participants

Assessment/Instruments

Procedures/Statistical analysis

Also, ethical considerations should also be indicated.

Authors: We rebuilt the Material and Method Section due to Reviewer`s suggestions. And ethical consideration is placed in the Institutional Review Board Statement Section.

Reviewer: It is interesting to have created a questionnaire by the researchers, however, it would have been interesting to have also administered some other standardized questionnaire to have a more scientifically consistent assessment, for example, the evaluation of anxiety, depression, stress or other another variable.

Authors: We developed it as a limitation of the study and a proposal for future studies in the discussion in the following way: “Instead, future research should include not only an specific questionnaire, but also completed the questionnaire results with specific depression or anxiety scales in order to reach powerful conclusions.”

Reviewer: Since a questionnaire has been created by the researchers, I suggest to include the complete questionnaire in the paper or explain better the parts of it in the method.

Authors: We have explained better in the method section as follow: An item-based questionnaire, Spanish and Polish version, was constructed following a comprehensive literature search. To maintain equality between the two questionnaires, both versions were back translated by a translator and subsequently reviewed by the research team. In this way, the accuracy of the translation was verified. The questionnaire consisted of 23 closed-ended, single- or multiple-choice questions. The questions included in the research tool concerned the activities of respondents during the pandemic, changes in working time, professional tasks performed, interpersonal relations and issues related to the area of mental functioning and health understood in a broad sense. At the end of the questionnaire, there was a data sheet containing basic socio-demographic data.

Reviewer: It is important to specify more precisely the assessment period, including the years, in order to know more precisely the moment of the time of the evaluation.

Authors: We have explain in Method section as follow: The survey was open from March 2021 to June 2022”

Reviewer: Try to include some more recent references regarding this field.

Author: We added references due to Reviewer`s suggestion.

Reviewer 2 Report

This is a potentially important report because studies are almost nonexistent on the pandemic’s impact on second-line providers who assist directly impacted persons (e.g., front-line staff) and families who are both primary and secondary victims (e.g., bereaved due to deaths; separated from social networks due to lock downs and burdened with childcare demands, economic hardships, and work pressures or job loss). 

However the use of the data to compare responses from the two countries yields few interpretable results because the differences may be due to factors unrelated to nationality that were not assessed (e.g., type of social service program and clientele; extent of direct and indirect exposure to covid and related suffering and deaths, personally and professionally), assessed but not included in the analyses (e.g., profession, gender, age), or simply due to the self-selected convenience samples.

Results that could be meaningful would require hypothesis-based analyses testing relationships between assessed variables reflecting pandemic-related changes in work and pandemic-related stressors with mental health outcomes (e.g., somatization, mood, social relations), and whether these were moderated by nationality, profession, gender or age.

Description also is needed of what "social services" were provided by these agencies (seems more like rehab given the large number of OTs, nurses, and "carers" and "others"--the latter two groups need to be defined more clearly, how agencies were selected, the response rate and how representative respondents were of the eligible workforce, and differences in all of these methodological factors between the 2 nations.

Author Response

Reviewer 2

Dear Reviewer,

We would like to thank you for a detailed review of our manuscript and all of your valuable remarks. We have addressed them all in detail below.

Reviewer: However the use of the data to compare responses from the two countries yields few interpretable results because the differences may be due to factors unrelated to nationality that were not assessed (e.g., type of social service program and clientele; extent of direct and indirect exposure to covid and related suffering and deaths, personally and professionally), assessed but not included in the analyses (e.g., profession, gender, age), or simply due to the self-selected convenience samples.

Authors: We agree with the reviewer's suggestion that the above-mentioned indicators were not included in the analyses. However, we would like to clarify that our project is ongoing and the suggested data will be collected and analyzed in subsequent studies.

Reviewer: Results that could be meaningful would require hypothesis-based analyses testing relationships between assessed variables reflecting pandemic-related changes in work and pandemic-related stressors with mental health outcomes (e.g., somatization, mood, social relations), and whether these were moderated by nationality, profession, gender or age.

Authors: Thank you for this helpful suggestion. At the current stage, the analyzes have already been carried out, but the project is ongoing and therefore, we will carry out all the analyzes suggested by the reviewer.

Reviewer: Description also is needed of what "social services" were provided by these agencies (seems more like rehab given the large number of OTs, nurses, and "carers" and "others"--the latter two groups need to be defined more clearly, how agencies were selected, the response rate and how representative respondents were of the eligible workforce, and differences in all of these methodological factors between the 2 nations.

Authors: We added proper paragraph in the Method section due to Reviewer`s suggestion. We want to explain that „carers” is the name of the profession that provides care and support services to the ward. And “others” mean instructors, art therapists and music therapists.

Reviewer 3 Report

The study analyzes if any differences existed between social welfare workers in Poland vs Spain in reported attitude about health and the effects of COVID 19. While explaining burnout in social welfare is an important issue importantly during the COVID-19 era, this study has major shortcomings. The shortcomings of the article are that the authors do not use an accepted research format, a scientific logic (does not have hypotheses for their analyses), appropriate analyses (uses item by item analyses), and confounds (do not discuss COVID-19 crisis).

After

“As there are no reports in the available literature on the negative effects of  the COVID-19 pandemic in this professional group, the aim of this study was to assess the  psychosocial and health effects of the COVID-19 pandemic experienced by staff employed  in social welfare institutions in Poland and Spain. “

Please insert hypotheses. Even there are no report some hypotheses are to be expected based on similar literature.

Materials

Questionnaire: The research tool was the authors’ questionnaire 

Please replace with The questionnaire was built for the present study by the authors.

The first examined correlation was that between the respondents’ country of residence and their attitude to their own health. There were only two statistically significant correlations between the variables studied.

Please replace the term correlations with the term differences. Please state the hypothesis and how the results do or do not support the hypotheses

Line 95 the need to modify particular forms of job-related activity due to COVID-19. The dif

Change with the reported or the perceived need. Similarly do throught the entire article

Line 102 The respondents were also asked what job-related activities they had to change due  to the COVID-19 pandemic. Statistically significant correlations were more often indicated by employees from Spain in the areas of remote work,

Please change to make sense. First it is not about correlations but differences, second are reports or perceptions [a reported increase or decrease in job-related duties and problems with self-acceptance –]

Third, please at limitations include the sample limitations such as self-selection. Also add that differnces limitations regarding the influence of vaccination status and trust in vaccines. For instance the authors may include at limitations Another limit is not considering the differences in the attitudes about vaccination of workers form Spain vs Poland. There are significant differences in attitudes about vaccination (Bell et al., 2022; Morar et al., 2022) that may explain the differences in mood deterioration. Further studies should investigate the attitudes about vaccination and vaccination status as well.

Bell S, Clarke RM, Ismail SA, Ojo-Aromokudu O, Naqvi H, Coghill Y, Donovan H, Letley L, Paterson P, Mounier-Jack S. COVID-19 vaccination beliefs, attitudes, and behaviours among health and social care workers in the UK: A mixed-methods study. PLoS One. 2022 Jan 24;17(1):e0260949. doi: 10.1371/journal.pone.0260949. PMID: 35073312; PMCID: PMC8786153.

Morar C, Tiba A, Jovanovic T, Valjarević A, Ripp M, Vujičić MD, Stankov U, Basarin B, Ratković R, Popović M, Nagy G, Boros L, Lukić T. Supporting Tourism by Assessing the Predictors of COVID-19 Vaccination for Travel Reasons. Int J Environ Res Public Health. 2022 Jan 14;19(2):918. doi: 10.3390/ijerph19020918. PMID: 35055740; PMCID: PMC8775532.

Author Response

Reviewer 3

Dear Reviewer,

We would like to thank you for a detailed review of our manuscript and all of your valuable remarks. We have addressed them all in detail below.

Reviewer: Please insert hypotheses. Even there are no report some hypotheses are to be expected based on similar literature.

Authors: We added the hypotheses in the Introduction section due to Reviewer`s suggestion.

Reviewer: “The research tool was the authors’ questionnaire“. Please replace with “The questionnaire was built for the present study by the authors.”

Authors: we change the sentence due to Reviewr`s suggestion.

Reviewer: „The first examined correlation was that between the respondents’ country of residence and their attitude to their own health. There were only two statistically significant correlations between the variables studied.” Please replace the term correlations with the term differences. Please state the hypothesis and how the results do or do not support the hypotheses.

Authors: We replaces word „correlation” into „difference” due to Reviewer`s suggestion in whole Result section. We also steted the hypotheses (Introduction) and explain if the results do or do not suport the hypotheses in the Discussion section.

Reviewer: Line 95 the need to modify particular forms of job-related activity due to COVID-19. The dif Change with the reported or the perceived need. Similarly do throught the entire article.

Authors: We changed it in Result and Discussion sections.

Reviewer: Line 102 “The respondents were also asked what job-related activities they had to change due  to the COVID-19 pandemic. Statistically significant correlations were more often indicated by employees from Spain in the areas of remote work” Please change to make sense. First it is not about correlations but differences, second are reports or perceptions [a reported increase or decrease in job-related duties and problems with self-acceptance –]

Authors: We changed the sentence due to Reviwer`s suggestion.

Reviewer: Third, please at limitations include the sample limitations such as self-selection. Also add that differnces limitations regarding the influence of vaccination status and trust in vaccines. For instance the authors may include at limitations Another limit is not considering the differences in the attitudes about vaccination of workers form Spain vs Poland. There are significant differences in attitudes about vaccination [] that may explain the differences in mood deterioration. Further studies should investigate the attitudes about vaccination and vaccination status as well.

Autors: We added the paragraph to the Study limitation section dut to Reviewer`s suggestion.

Round 2

Reviewer 2 Report

Additions to the text have improved the paper but some more are still needed:  
  1. The description of yhe worker categories in the response to R 2 should be added to the paper
  2. Justification for the hypotheses is needed in the introduction (some that are provided in the Discussion could be added here)
  3. The limitation of different age cohorts (younger in Spain) and higher education (more respondents from Spain with higher education) should be emphasized as a possible explanation of the national differences
  4. The relativele few men surveyed should be added as a limitation

Author Response

Reviewer 2

Dear Reviewer,

We would like to thank you for a detailed review of our manuscript and all of your valuable remarks. We have addressed them all in detail below.

Reviewer: The description of yhe worker categories in the response to R 2 should be added to the paper

Authors: We added the description of the workers to the paper due to Reviewer`s suggestion

Reviewer: Justification for the hypotheses is needed in the introduction (some that are provided in the Discussion could be added here)

Authors: We added appropriate sentence in the Introduction section

Reviewer: The limitation of different age cohorts (younger in Spain) and higher education (more respondents from Spain with higher education) should be emphasized as a possible explanation of the national differences

Authors: We added appropriate sentence to the Limitation section due to Reviewer`s suggestion

Reviewer: The relativele few men surveyed should be added as a limitation

Authors: We added appropriate sentence to the Limitation section due to Reviewer`s suggestion

Reviewer 3 Report

“This was 25 coronavirus 2019 (COVID-19)” please add coronavirus disease 2019

The psychological effects of covid 19  please write with capital letters COVID-19

Spaniards have more concerns about their own health than Poles

Please replace with Social workers from Spain…. than social workers from Poland

(they may be also foreign workers and the terms Spaniards may not be appropriate)

I would suggest to consider this suggestion in the entire article

Line 102 please change correlation

Author Response

Reviewer 3

Dear Reviewer,

We would like to thank you for a detailed review of our manuscript and all of your valuable remarks. We have addressed them all in detail below.

Reviewer: “This was 25 coronavirus 2019 (COVID-19)” please add coronavirus disease 2019

Authors: We added the word due to Reviewer`s suggestion

Reviewer: The psychological effects of covid 19  please write with capital letters COVID-19

Authors: We wrote the word with the capital letters due to Reviewer`s suggestion

Reviewer: Spaniards have more concerns about their own health than Poles. Please replace with Social workers from Spain…. than social workers from Poland (they may be also foreign workers and the terms Spaniards may not be appropriate). I would suggest to consider this suggestion in the entire article

Authors: We changed the words „Poles” and „Spaniards” into „employees from Poland” and „employees from Spain” in whole manuscript due to Reviewer`s suggestion

Reviewer: Line 102 please change correlation

Authors: We changed word „correlation” to „differences” due to Reviewer`s suggestion